# Unravelling the Inflammatory Processes in the Early Stages of Diabetic Nephropathy and the Potential Effect of (*S_s_*)-DS-ONJ

**DOI:** 10.3390/ijms23158450

**Published:** 2022-07-30

**Authors:** Laura Gómez-Jaramillo, Fátima Cano-Cano, Elena M. Sánchez-Fernández, Carmen Ortiz Mellet, José M. García-Fernández, Martín Alcalá, Fabiola Álvarez-Gallego, Marta Iturregui, María del Carmen González-Montelongo, Antonio Campos-Caro, Ana I. Arroba, Manuel Aguilar-Diosdado

**Affiliations:** 1Unidad de Investigación, Instituto de Investigación e Innovación en Ciencias Biomédicas de la Provincia de Cádiz (INiBICA), 11009 Cádiz, Spain; canocano.fatima@gmail.com (F.C.-C.); m.iturregui@gmail.com (M.I.); mcarmen.gonzalez@inibica.es (M.d.C.G.-M.); antonio.campos@uca.es (A.C.-C.); manuel.aguilar.sspa@juntadeandalucia.es (M.A.-D.); 2Departamento de Endocrinología y Nutrición, Hospital Universitario Puerta del Mar, Universidad de Cádiz, 11009 Cádiz, Spain; 3Departamento de Química Orgánica, Facultad de Química, Universidad de Sevilla, 41012 Sevilla, Spain; esanchez4@us.es (E.M.S.-F.); mellet@us.es (C.O.M.); 4Instituto de Investigaciones Químicas (IIQ), CSIC—Universidad de Sevilla, 41092 Sevilla, Spain; jogarcia@iiq.csic.es; 5Departamento de Química y Bioquímica, Facultad de Farmacia, Universidad San Pablo-CEU, CEU Universities, 28668 Boadilla del Monte, Spain; martin.alcaladiazmor@ceu.es (M.A.); f.alvarez17@usp.ceu.es (F.Á.-G.); 6Departamento de Endocrinología y Nutrición, Hospital Universitario Jerez de la Frontera, Universidad de Cádiz, 11407 Cádiz, Spain; 7Área de Genética, Departamento de Biomedicina, Biotecnología y Salud Pública, Universidad de Cádiz, 11003 Cádiz, Spain

**Keywords:** inflammation, diabetic nephropathy, epithelial mesenchymal transition, autophagy, (*S_s_*)-DS-ONJ

## Abstract

Inflammatory processes play a central role in the pathogenesis of diabetic nephropathy (DN) in the early stages of the disease. The authors demonstrate that the glycolipid mimetic (*S_s_*)-DS-ONJ is able to abolish inflammation via the induction of autophagy flux and provokes the inhibition of inflammasome complex in ex vivo and in vitro models, using adult kidney explants from BB rats. The contribution of (*S_s_*)-DS-ONJ to reducing inflammatory events is mediated by the inhibition of classical stress kinase pathways and the blocking of inflammasome complex activation. The (*S_s_)*-DS-ONJ treatment is able to inhibit the epithelial-to-mesenchymal transition (EMT) progression, but only when the IL18 levels are reduced by the treatment. These findings suggest that (*S_s_*)-DS-ONJ could be a novel, and multifactorial treatment for DN.

## 1. Introduction

Type 1 diabetes mellitus (T1DM) is a chronic disease that is associated with the development of complications including retinopathy, nephropathy, neuropathy and cardiovascular morbidity. It is recognized that chronic low-grade inflammation and the activation of the innate immune system are closely involved in the pathogenesis of T1DM [1]. An improved understanding of the mechanisms linking inflammation to diabetes and related complications has increased the interest in targeting inflammatory pathways as part of the strategy to prevent or control disease progression [2]. Diabetic nephropathy (DN) is a multifactorial chronic renal dysfunction that represents a significant risk of morbidity and mortality within all patient populations with diabetes. It is considered a microvascular complication and occurs in both type 1 diabetes mellitus (T1DM) and type 2 diabetes mellitus (T2DM). DN originates from the damage or danger signals released by injured renal cells, which can trigger remodeling processes by stimulating renal cells and activating the immune cells of both the innate and adaptive systems [3,4]. Even though cytokines are generally thought to be released from activated immune cells, renal cells can likewise secrete pro-inflammatory cytokines and others metabolites that may exacerbate DN [3]. Today, it is known that renal tubular cells are able to produce interleukin 1β (IL-1β), interleukin 18 (IL-18) and tumor necrosis factor-alpha (TNF-α) [5,6,7] which, with diverse actions, are potentially involved in the development and complications of DN [8,9,10,11,12].

Although the inflammasome was originally believed to be specific to the innate immune system, its activation and signaling pathways in non-immune cells have recently been described [13]. The release of the pro-inflammatory cytokines IL-1β and IL-18 is controlled by the inflammasome. Its expression has been demonstrated in the proximal tubules of renal cells [14], thereby promoting DN progression [15]. Caspase-1, an essential component of the inflammasome complex, initiates a pro-inflammatory response through the cleavage, and thus the activation, of IL-1β and IL-18, inducing a downstream inflammatory response in renal cells [11,16].

Studies have indicated that DN is associated with decreased autophagy activity [17,18]. Therefore, it is not surprising that autophagy has gained considerable attention in recent years as the possible pathological basis of DN [19]. Autophagy could be interpreted as an adaptive response that might confer protection against persistent inflammation [20].

It has been widely demonstrated that the Epithelial-to-Mesenchymal Transition (EMT) is a direct contributor to the development of renal fibrosis in DN [21]. In the early inflammatory phase of renal fibrosis, cytokines produced by infiltrating local cells play a decisive role in initiating the phenotypic change in the EMT, mainly involving proximal tubular cells and podocytes [22]. During EMT, epithelial cells lose cell–cell contact, which allows for the switch to a mesenchymal phenotype [23]. The phenotypic changes are associated with decreased expression of epithelial markers and enhanced expression of mesenchymal markers, such as E-cadherin and fibronectin, respectively [24], as well as proteins with de novo acquisition such as α-smooth muscle actin (α-SMA) [25]. Different EMT transcription factors (EMT-TF) such as Snail and *ZEB-1* are often co-expressed in various combinations and intervene to orchestrate complex EMT programs depending on the specific biological context [26].

(*S_S_*)-(DS-ONJ) belongs to a family of compounds called sulfur-linked sp^2^-iminosugar glycolipids (sp^2^-IGLs). In contrast to other immunoregulatory glycolipids, the sp^2^-IGLs are metabolically stable and can be prepared in pure anomeric form with total stereoselectivity, thereby providing a suitable platform for glycodrug design [27,28,29,30,31,32]. Increasing evidence suggests that sp^2^-IGLs may offer anti-inflammatory protection against diabetic complications [33]. Our group has recently demonstrated the anti-inflammatory effect of the (*S_S_*)-DS-ONJ compound on neuro-immunomodulation during Diabetic Retinopathy (DR) progression [34]. Given that inflammation is one of the first mechanisms underlying DN, compounds capable of modulating inflammation could be considered powerful candidates for the treatment of DN.

Our aim in this study was to decipher the different processes that occur in the early stage of DN and test the potential therapeutic effect of the (*S_S_*)-DS-ONJ molecule in the context of the DN.

## 2. Results

### 2.1. (S_S_)-DS-ONJ Treatment Reduces the Pro-Inflammatory Response in Mouse Cortical Ttbular Cells from Kidney (MCT) Cells under Diabetic Inflammatory Stimulus by Autophagy Flux Induction

The MCT mouse cortical tubular epithelial cell line reproduced the inflammatory features of kidney failure when they were cultured in a pro-inflammatory environment using a cocktail of cytokines. In order to determine the possible cytotoxic effects of (*S_S_*)-DS-ONJ (see chemical structure in Figure 1A) on MCT cells, violet crystal staining was used to measure MCT cell viability after incubation with increased concentrations of (*S_S_*)-DS-ONJ (0.1–50 µM). As indicated in Figure 1B, the effect was dose-dependent, showing a reduction of 50% of cell viability at the highest dose of the compound (50 µM).

The low non-cytotoxic concentrations (1–10 µM) were checked for the anti-inflammatory activity of (*S_S_*)-DS-ONJ using the significant reduction of CKs treatment-induced nitrites production and iNOS levels (Figure 1C and Figure 2A). The working concentration of (*S_S_*)-DS-ONJ was established at 10 µM for further experiments.

The robust increase in iNOS levels observed after CKs-induced stimulus was strongly inhibited upon (*S_S_*)-DS-ONJ treatment, likely as a direct result of the treatment (Figure 2A). Similarly, *Tnfa* and *Il1b* expression were also significantly downregulated in the presence of (*S_S_*)-DS-ONJ (Figure 2B).

Due to the chronic inflammatory processes that occurring in the DN disease, we explored a possible effect of (*S_S_*)-DS-ONJ on inflammasome activation. We have detected a marked increase in procaspase-1 levels and *Nlrp3* expression in response to CKs (Figure 2C, D), which were significantly reverted with (*S_S_*)-DS-ONJ treatment.

Autophagy has been considered a key player in the resolution of inflammatory processes [35,36]. As can be seen in Figure 2E, the CKs stimulation blocked the autophagic flux, with the accumulation of p62 levels and the LC3-II/I ratio being significantly decreased. Treatment with (*S_S_*)-DS-ONJ upregulated the activation of LC3-II/I related to re-establish autophagic flux but did not restore the p62 levels to their basal levels. In order to determine the potential point where (*S_S_*)-DS-ONJ modulated the autophagy flux, MCT cells were treated with chloroquine (CLQ), an inhibitor acting at a late stage of the process, specifically the degradation of autophagolysosomes, that promotes an increase in the LC3 II/I ratio and p62 levels. The presence of both CLQ and (*S_S_*)-DS-ONJ caused a significant increase in the LC3II/I ratio and p62 accumulation compared to CKs and (*S_S_*)-DS-ONJ condition (Figure 2F). The (*S_S_*)-DS-ONJ treatment did not modulate the late inhibition in the autophagic flux. However, (*S_S_*)-DS-ONJ treatment induces the LC3 lipidation and partially promotes the restoration of p62 levels as a result of inhibiting inflammatory pathways.

### 2.2. Stress Kinases Pathways Are Modulated by (S_s_)-DS-ONJ in MCT Cells under Diabetic Conditions

The classical stress kinases pathways and NFκB-mediated signaling pathways involved in the inflammatory processes were examined. CKs-stimulation induced a maximal effect at 30 min and at 90 min in the phosphorylation of JNK (Figure 3A) and p38α MAPK, respectively (Figure 3B). However, (*S_S_*)-DS-ONJ treatment showed an anti-inflammatory effect by inducing a decrease in the phosphorylation of both kinases (Figure 3A,B) and, moreover, prevented CK-mediated nuclear translocation of p65-NF-κB (Figure 3C,D). These results indicate that the compound may have anti-inflammatory effects. The potent inhibitory effect of (*S_S_*)-DS-ONJ is stronger on the JNK than on the p38α MAPK pathway, with a significant phosphorylation reduction in CK-induced phosphorylation at 15 min and at 30 min, respectively. The p38α MAPK inhibitor SB203580 was next used for determining the actions of (*S_S_*)-DS-ONJ on this signaling pathway. As can be seen in Figure 3E, a substantial reduction in iNOS protein expression consistent with the inhibition of MAPK signaling was observed. Protein levels of iNOS were also reduced by the combination of CKs, SB203580 and (*S_S_*)-DS-ONJ. No synergic effects were detected, however, reflecting the independent behavior of SB203580 and (*S_S_*)-DS-ONJ (Figure 3E).

### 2.3. MCT Cells Cultured in a Diabetic Environment Show EMT Processes Mediated by IL18

The EMT phenomenon may be observed in the inflammation process in DN. A decrease in the expression of an epithelial marker, E-cadherin, and the up-regulation of the expression of a mesenchymal marker, such as α-SMA [22], define the classical activation of EMT. In MCTs cells stimulated with CKs, the inflammatory environment induced the E-cadherin down-regulation; however, (*S_S_*)-DS-ONJ treatment was not able to recover the basal condition (Figure 4A). No other changes in α-SMA protein levels were detected, but an increase in the mRNA expression of *Acta-2 (*α-SMA *gene*) was observed in CKs-stimulated MCT cells. This is in agreement with a representative induction of EMT at the mRNA levels, which was significantly reduced in the presence of (*S_S_*)-DS-ONJ (Figure 4B). Tubular epithelial cells are susceptible to TGF-β1, which is produced by both the injured tubule and the infiltrating cells, and instigates a fibrogenic program that includes EMT [21]. Looking for a corroborative feature of our previous results, *Tgfb1* expression was analyzed and was found to be increased upon stimulation with CKs, but not reduced in the presence of (*S_S_*)-DS-ONJ (Figure 4C).

Recent evidence has demonstrated that IL18 promotes EMT and that the neutralization of IL18 in mice suppresses this transition. The inhibition of IL18 in these mice leads to a decrease in α-SMA and an increase in E-cadherin levels [37]. *Il18* expression was increased in MCTs upon stimulation with CKs-stimulation and the (*S_S_*)-DS-ONJ treatment was not able to decrease this expression (Figure 4C). However, when the expression of *Il18* was reduced with specific small interference RNA (siRNA), the EMT process was reversed in CKs-stimulated MCTs. This can be inferred from the increase in the expression of *Cadherin-1* and the decrease in *Acta-2* expression, but there was no a significant effect on the expression of *Tgfb1* (Figure 4D). Treatment with a scramble sequence (*SC*) did not interfere with the gene expressions analyzed. (*S_S_*)-DS-ONJ’s actions on EMT are contingent on the presence of *Il18*.

Regarding the identification of specific transcriptional factors involved in the EMT process that are modulated by (*S_S_*)-DS-ONJ treatment, *ZEB-1,* Snail and Slug were analyzed as potential transcription factors involved in the onset of the EMT. *ZEB-1* levels were increased by CKs stimuli and modulated to basal levels in the presence of (*S_S_*)-DS-ONJ, but no changes were observed in Snail or Slug levels (Figure 4E).

### 2.4. AK Explant Cultures from Non-Diabetic Rats Stimulated with CKs Reproduce the Molecular Events Observed in MCT Cells in the Diabetic Context and the (S_S_)-DS-ONJ Treatment Protects from Pro-Inflammatory Insults

As has previously been shown, the stimulation with CKs promoted the increment of iNOS levels and its mRNA expression (Figure 2A), as well as the expression levels of pro-inflammatory cytokines, such as *Tnfa* and *Il1b* (Figure 2B).

Ex vivo assays were performed to gain insights into the potent anti-inflammatory effect of (*S_S_*)-DS-ONJ on AK explants from Wistar rats stimulated with CKs. CKs treatment causes an increase in iNOS levels and its mRNA expression (Figure 5A), as well as the expression of the mRNA pro-inflammatory cytokines, such as *Tnfa* and *Il1b* (Figure 5B). The co-treatment of AK explant with CKs and (*Ss*)-DS-ONJ showed that this compound is capable of reducing the levels of expression of these pro-inflammatory mediators (Figure 5A, B). Furthermore, an analysis of *Nlrp3* indicates a modulation of inflammasome complex activation due to (*S_S_*)-DS-ONJ’s effects (Figure 5B) via increase in the *Nlrp3* expression under CKs treatment that is significantly reduced in (*S_S_*)-DS-ONJ treatment.

In order to get more insights into the mechanisms involved in the potential protective effects of (*S_S_*)-DS-ONJ in AK explants, we tested the autophagy flux and we found that the combined CKs plus (*S_S_*)-DS-ONJ markedly increased the expression of LC3-II/I but did not restore the p62 expression (Figure 5C) in CKs-stimulated AK explants.

Similarly, to what was observed in MCT cells, the treatment of AK explants with CKs led to a significant reduction in E-cadherin levels and the up-regulation of α-SMA. These are indicative of EMT progression, which was significantly reverted in the presence of (*S_S_*)-DS-ONJ in protein according to a histological examination, as were the mRNA levels (Figure 6A–C). Furthermore, a reduction in *Il18* expression in the presence of (*S_S_*)-DS-ONJ was observed (Figure 6D). The expression levels of *Tgfb1* were not changed in the presence of CKs nor by (*S_S_*)-DS-ONJ treatment (Figure 6C). The ex vivo system showed the modulation of molecules implicated in the classical features of EMT, despite the findings with the in vitro system. The tissular context in the EMT response may be critical for fully reproducing the signaling pathways involved in the process.

### 2.5. (S_S_)-DS-ONJ Abolishes Classic Hallmarks of DN in AK Explants from BB Rats

The BB rat develops diabetes spontaneously at 11 weeks old; however, it shows inflammatory progress from 7 weeks of age [38]. Recently, we have reported that BB rats present important inflammatory events prior to hyperglycemia or preclinical diabetic status [39]. As is shown in Figure 7A, AK explants from BB rats at 7 weeks old, with euglycemic values in the blood, present increased levels of iNOS protein and mRNA expression that were significantly reduced in the presence of (*S_S_*)-DS-ONJ.

Moreover, (*S_S_*)-DS-ONJ treatment induces a decrease in the mRNA expression of pro-inflammatory cytokines such as *Tnfa*, *Il1b* and the inflammasome component *Nlrp3* (Figure 7B). Furthermore, (*S_S_*)-DS-ONJ activates the autophagy flow in the AK explants of diabetic rats, as can be seen in the increased LC3II/I ratio (Figure 7C).

As was the case in the in vitro system, the (*S_S_*)-DS-ONJ treatment in AK explants from BB rats, did not modulate the E-cadherin protein levels in AK explants from BB (Figure 8A–C). Immunohistochemical analyses also demonstrated no changes in E-cadherin protein levels but a decreased in α-SMA protein levels (Figure 8B). Thus, (*S_S_*)-DS-ONJ did not reverse the EMT process despite the observed decrease in α-SMA protein levels and *Tgfb1* expression *(*Figure 8A–D). The treatment likely stopped the EMT progression. The critical point in EMT reversion was the reduction in *Il18* expression and the (*S_S_*)-DS-ONJ treatment did not reduce *Il18* expression. An analysis of the specific transcriptional factors involved in EMT showed that, (*S_S_*)-DS-ONJ reduced the protein levels of ZEB-1 but increased those of Snail (Figure 8E) in BB AK explants.

## 3. Discussion

Different processes occur during the early stages of DN that contribute to the development of the classic pathophysiology associated with T1DM. Inflammation, autophagy dysfunction and the EMT trigger renal failure, and the possible use of a new treatment limiting their impact during the first stages of DN could contribute to delaying disease progression, resulting in a better response over time in the current management of DN.

DN is considered a secondary diabetes complication. However, it is noteworthy that this work shows an incipient kidney inflammation prior to the development of hyperglycemia. In this way, it has been established that before suffering elevated blood glucose levels due to pancreatic failure, the kidney is being affected by the inflammatory phenomenon. The characterization of MCT cells as an in vitro model for studying the early events of DN has contributed to dissecting the signaling pathways involved in this pathology and the possible protective role of (*S_S_*)-DS-ONJ on DN progression [39]. Further, the use of AK explants as a pathophysiological ex vivo model in DN has been relevant to testing therapeutic approaches.

Previous reports demonstrate a typical inflammatory response in the MCT cell line upon exposure of the cell culture to individual pro-inflammatory cytokines [40,41,42]. In the present work, MCT cells and AK explants from WT rats were cultured under a mixture of pro-inflammatory cytokines in order to mimic the pro-inflammatory T1DM-associated environment. An inflammatory profile similar to the in vivo physiological context, i.e., a low-grade inflammation environment, was thus obtained. In both the in vitro and the ex vivo systems, CKs stimulation induced a strong nitrite production supported by iNOS induction. Regarding inflammation resolution, there is strong evidence that autophagy is directly involved [36]. The inhibition of autophagy correlates with increased levels of p62. Changes in this protein can be cell type and context- specific. In some cell types, there is no change in the overall amount of p62 despite strong levels of autophagy induction. In other contexts, a robust loss of p62 does not correlate with increased autophagic flux [17]. CLQ is an inhibitor of autophagic flow in the late stages at the fusion point of the phagosome with the lysosome [43] so that the LC3 II that is part of the autophagolysosome cannot be degraded, leading to the accumulation of LC3II after treatment with CLQ. According to our results, (*S_S_*)-DS-ONJ treatment restores the levels of LC3 II, probably acting on the early stages of the autophagic pathway. In the experimental systems used, namely MCT cells and AK explants from WT rats treated with CKs, a pro-inflammatory environment inhibited the autophagic flow. Autophagy was restored by (*S_S_*)-DS-ONJ pre-treatment since LC3 levels were equalized to basal levels, which could induce the resolution of the incipient inflammatory process associated with DN. This inflammatory response of MCT cells and AK explants was straightened by inflammasome complex activation. The pivotal role of the inflammasome complex during DN has recently been demonstrated [14]. Once the protein complexes are formed, the inflammasome activates caspase 1, which proteolytically activates the pro-inflammatory cytokines IL-1β [44] and IL-18 [45]. In MCT cells and AK explants cultured under CKs conditions, an accumulative expression of procaspase-1 and Nlrp3 were observed as indicators of inflammasome activation. However, (*S_S_*)-DS-ONJ administration reduced and promoted the inhibition of *Tnfa* and *Il1b mRNA* expression. Previous studies have reported reductions in tubular injury, inflammation and fibrosis associated with reduced caspase-1 activation, as well as the failure of IL-1β and IL-18 maturation in kidney-specific NLRP3 knockout mice [46]. In the same way, we detected a clear decrease in *Il18* expression under (*S_S_*)-DS-ONJ treatment, which could be related to inflammasome inhibition.

The mechanisms involved in the modulation of the above processes were studied next. To this end, the differential participation of the classic stress kinase pathways in the inflammatory pathway activation and regulation by (*S_S_*)-DS-ONJ treatment was assessed. The inflammatory response in different pathologies is highly linked to specific intrinsic pathways that orchestrate the pro-inflammatory microenvironment, and IL-1β has been associated with this process [47,48]. In recent studies, IL-1β has been targeted as an essential activator for NF-κB-regulated gene activation, which includes cytokines and chemokines required for the establishment of a diabetogenic environment. The activation of JNK by phosphorylation under CKs stimulus and its inhibition by (*S_S_*)-DS-ONJ was faster than p38α-MAPK activation in MCTs cells. The activation of JNK signaling in tubular epithelial cells could contribute to the progression of chronic kidney disease [49] in DN. In this regard, the potent inhibition by (*S_S_)*-DS-ONJ delayed the inflammatory process in our system. The use of SB203580 in the presence of CKs and (*S_S_*)-DS-ONJ reveals that (*S_S_*)-DS-ONJ and CKs potentially compete to bind to the same site in the p38α MAPK protein. Thus, the anti-inflammatory effects of (*S_S_*)-DS-ONJ were partially abolished in the presence of the classical inhibitor of p38α MAPK activation.

The JNK pathway interacts with the pro-fibrotic signaling led by TGF-β1, where JNK activation can augment TGF-β1 gene transcription and induce the expression of enzymes that activate the latent form of TGF-β1, and JNK directly enhances the transcription of pro-fibrotic molecules [50,51]. As our results show, we have detected an increase in *Tgfb*1 expression caused by CKs stimulation in cultured MCT cells. However, *Tfgb1* mRNA levels were not modulated in AK explants from WT rats under CKs-stimulation conditions. The weak effect on *Tgfb1* expression by CKs stimulation in MCT cells could be due to the low grade inflammation environment that induces the diabetic milieu and the short exposure time. Moreover, in the ex vivo approach, we detected a differential response to inflammatory induction that depends on the inflammatory initial status of AK explants.

The anti-inflammatory role of (*S_S_*)-DS-ONJ has been detected in other pathological complications associated with T1DM, such as diabetic retinopathy [34]. The EMT represents an underlying inflammatory process that concurs with the development of fibrosis, as the final step of EMT and renal failure. New therapies have focused on the modulation or delaying of EMT and blocking fibrosis associated with DN progression [52]. In a Streptozotocin (STZ)-induced model of diabetes in rats, the vascular changes, density of α-SMA-positive cells, extensively vacuolated cells, inflammatory cells infiltrated and tubular interstitial fibrosis are evident in the diabetic kidney [53] indicating that the model rats held obvious pathological damage and developed renal fibrosis. As occurs with other treatments, the renal injury and fibrosis in STZ-induced diabetic rats were ameliorated via modulating the protein expressions of downstream inflammatory factors to protect the kidney in STZ-induced model of diabetes in rats [54]. Treatment with (*S_S_*)-DS-ONJ could mediate the progression of EMT, even though it is not able to reverse the process in in vitro settings and in BB AK explants. However, in WT AK explants the (*S_S_*)-DS-ONJ treatment restores E-cadherin and α-SMA levels. Probably, IL18 modulation is involved in these differential responses. The recent consensus in EMT processes defines the specific transcription factor involved in the physiological or pathological contexts. In this regard, the increased E-cadherin expression was associated with decreased expression of *ZEB1* and *ZEB2*. This transcriptional repression is responsible for protecting tubular epithelial cells from the mesenchymal transition [55,56]. Additionally, (*S_S_*)-DS-ONJ has a potent effect on WT AK explants treated with CKs, and reverses EMT at basal levels. Our results further show the inhibition of classical features of EMT by (*S_S_*)-DS-ONJ treatment in the ex vivo model of DN consisting of AK explants stimulated with inflammatory CKs. The protective effects of (*S_S_*)-DS-ONJ correlates with the modulation of E-cadherin and α-SMA as key players in the EMT, and the down-regulation of *Il18*. We have demonstrated that IL-18 contributes in this regard because the knockdown of IL-18 suppressed the EMT in MCT cells.

Altogether, the events modulated by (*S_S_*)-DS-ONJ trigger a reduction in deleterious processes associated with DN: inflammation, inflammasome complex activation and EMT progression. We have analyzed the potential effect of (*S_S_*)-DS-ONJ as a promising therapeutic treatment in DN due to its potent anti-inflammatory and anti-fibrotic effect (see Graphical abstract). The modulation of different processes that occur in the initial steps or during DN progression could reduce the adverse effects of the disease. AK explants from BB rats, an animal model for the study of DN [35], significantly reduced the levels of the pro-inflammatory parameters in the presence of (*S_S_*)-DS-ONJ, compared to AK explants from BB rats in basal condition. Possible due to its involvement in inflammatory resolution, the autophagy flux is increased upon (*S_S_*)-DS-ONJ administration to AK explants from BB rats. Although (*S_S_*)-DS-ONJ does not reverse EMT to the basal condition, it slows the progression of EMT toward a healthier renal situation. In summation, the use of (*S_S_*)-DS-ONJ as a drug candidate with multiple therapeutic targets during DN could delay disease progression and reverse some of the most important DN hallmarks, such as inflammation and EMT induction.

## 4. Material and Methods

### 4.1. Reagents

Fetal bovine serum (FBS) and culture media were obtained from Invitrogen (Grand Island, NY, USA). Bovine serum albumin (BSA), crystal violet, glutaraldehyde, N-(1-naphthyl) ethylenediamine (NEDA), sulfanilamide, bacterial lipopolysaccharide (LPS), SB203580 and chloroquine were purchased from Sigma-Aldrich (St. Louis, MO, USA). Acrylamide and immunoblot PVDF membranes were purchased from Bio-Rad (Madrid, Spain). BCA reagent and chemiluminescent HRP substrate were purchased from Pierce (Rockford, IL, USA). Cytokines such as TNFα, IL1β and IFNγ were obtained from Prepotech (Rocky Hill, NJ, USA). Protease inhibitors cOmplete-EDTA free were obtained from Roche (Germany). Total RNA was extracted with Trizol^®^ reagent (Invitrogen, Madrid, Spain) and reverse transcribed using the iScript gDNA Clear cDNA Synthesis Kit from BioRad (Madrid, Spain). qPCR was performed with iTaq Universal probes Supermix from BioRad (Madrid, Spain) in a CFX Connect Real-Time System from BioRad (Madrid, Spain). Thiobarbital was obtained from Braun Medical, S.A. (Rubí, Barcelona, Spain).

### 4.2. Antibodies

Appendix A shows the antibodies used for Western-blot and immunofluorescence assays. The protein signals were analyzed using aChemiDoc chemiluminescence device (Bio-Rad, Madrid).

### 4.3. Synthesis of the sp^2^-Iminosugar Glycolipid (S_S_)-DS-ONJ

(*S_S_*)-(1*R*)-1-Dodecylsulfinyl-5*N*,6*O*-oxomethylidenenojirimycin (referred to as (*S_S_*)-DS-ONJ, Figure 1A) was synthesized from (1*R*)-1-dodecylthio-5*N*,6*O*-oxomethylidenenojirimycin by using *m*-chloroperbenzoic acid as oxidant agent, following the procedure previously reported [57]. Subsequent purification by column chromatography allowed the separation from its (*S*_R_)-diastereomer (*S*_R_)-DS-ONJ.

### 4.4. Cell Culture

Mouse cortical tubular from kidney (MCT) cell line was supplied by Dr. Ana Belen Sanz (Madrid, Spain). 100,000 cells per well were seeded in a 6 multiwell plate (Sarstedt, Germany). The cultured conditions were 37 °C in a humidified atmosphere with 5% CO_2_ in RPMI supplemented with 10% heat-inactivated FBS, 1% penicillin/streptomycin and 2 mM L-glutamine. During the experiment, cells were grown up to 70% confluence and then washed with PBS and further cultured in a serum-free medium.

#### 4.4.1. CKs Treatment

Cells were treated previously for 4 h with (*S_S_*)-DS-ONJ, and then MCT cells were stimulated with a cytokines cocktail (CKs; 30 ng/mL TNFα, 30 UI/mL IF-γ and 1 ng/mL IL1β), (*S_S_*)-DS-ONJ (10 μM), (*S_S_*)-DS-ONJ+CKs and/or vehicle for 24 h.

#### 4.4.2. Chloroquine Treatment

MCT cells were pretreated for 4 h with (*S_S_*)-DS-ONJ (10 μM) followed by incubation with 50 µM of chloroquine and/or CKs for 18 h.

#### 4.4.3. SB203580 Treatment

MCT cells were pretreated for 30 min with SB203580 (10 μM) followed by incubation with (*S_S_*)-DS-ONJ (10 μM) for 4 h. Then, cells were stimulated with CKs for 24 h.

### 4.5. Adult Kidney Explants

All animal procedures were performed with the approval of the Cádiz University School of Medicine (Cádiz, Spain) Committee for the Ethical Use and Care of Experimental Animals (*005_abr20_PI2—ITI-012-2019*). All animal experimentation followed the recommendations of the Federation of European Laboratory Animal Science Associations (FELASA).

Bio-Breeding (BB) and Wistar rats were kept under conventional conditions in an environment-controlled room (20–21 °C, 12 h light-dark cycle) with water and standard laboratory rat chow available ad libitum. Blood extracted from the tail vein was used in BB rats for weekly glucose measurements using an automatic glucose monitor (Freestyle Optium Neo, Abbott, Madrid, Spain).

We used 7-week-old BB rats, a classical animal model of T1DM, to study inflammatory processes in the early stages of T1DM in a pre-diabetic status with euglycemia. Ex vivo assays were performed with kidneys from male or female Wistar and BB rats.

Animals were euthanized by an overdose of anesthesia (sodium thiobarbital). The kidneys were extracted in PBS, and adipose tissue and the suprarenal glands were removed. The slices of 1 mm of cross-sections were cultivated individually in RPMI supplemented with 10% heat-inactivated FBS, 1% penicillin/streptomycin, 2 mM L-glutamine, 10 µg/mL of gentamycin (Laboratories Normon, Spain). The AK explants were cultured with CKs (60 ng/mL TNFα + 60 UI/mL IF-γ + 2 ng/mL IL1β), (*S_S_*)-DS-ONJ (20 µM) or (*S_S_*)-DS-ONJ + CKs and/or vehicle as indicated in the figure legends and were cultured at 37 °C in a humidified atmosphere with 5% CO_2_ for 24 h.

### 4.6. Analysis of the Cellular Viability by Crystal Violet Staining

After cell treatments, culture medium was discarded from plates, and the remaining viable adherent cells were fixed with 10% of glutaraldehyde in PBS. Then, cells were stained with crystal violet (0.1% *w*/*v* in water) for 20 min, rinsed with tap water and allowed to dry. After that, 10% of acetic acid in water was added to solubilize them. The absorbance of each plate was read spectrophotometrically at 590 nm (Power Wave, Biotek, Torino, Italy).

### 4.7. Analysis of Nitrites (NO_2_^−^)

Levels of NO_2_^−^ were measured by using the Griess method [58] and quantified by a colorimetric method at 548 nm in a microplate reader (Power Wave, Biotek, Torino, Italy).

### 4.8. Western Blot

Kidney slides or MCT cells were homogenized in lysis buffer containing 125 mM Tris-HCl, pH 6.9, 2% SDS, and 1 mM DTT supplemented with protease inhibitors. All debris was removed by centrifugation at 14,000× *g* for 10 min at 4 °C and protein concentration was quantified using the Bio-Rad protein assay with BSA as a standard. Equivalent amounts of protein were resolved using denaturing sodium dodecyl sulphate-polyacrylamide gel electrophoresis (SDS-PAGE), followed by transfer to PVDF membranes (Merck Millipore, Cork, IRL). Membranes were blocked using 5% nonfat dried milk or 3% BSA in 10 mM Tris-HCl, 150 mM NaCl, pH 7.5 (TBS), and incubated overnight with several primary antibodies (1:1000 unless otherwise stated) in 5% nonfat dried milk or 3% BSA in 10 mM Tris-HCl, 150 mM NaCl, pH 7.5 (TBS). Immunoreactive bands were visualized using the enhanced chemiluminescence reagent (Bio-Rad). The fold change relative to the basal condition is shown. Blots were quantified by performing scanning densitometry, and the results are mean ± S.E.M. Representative images are shown.

### 4.9. Immunofluorescence

MCT cells were seeded on coverslips for 24 h before adding CKs and/or (*S_s_*)-DS-ONJ in a serum-free medium. After treatment, cells were washed in PBS, fixed with 4% paraformaldehyde in PBS for 10 min at room temperature, washed in PBS and permeated with 0.4% Triton X-100 in PBS for 20 min. Non-specific binding was blocked in PBS containing 3% BSA and 0.1% Triton X-100 for 2 h and the cells were then left overnight in a humid chamber at 4 °C with rabbit anti-p65 NFκB antibody in blocking buffer (TBS containing 3% BSA and 1% Triton X-100). Subsequently, the cells were washed and incubated in the dark for 2 h with an anti-rabbit conjugated Alexa 488 antibody (Molecular Probes, ThermoFisher Scientific, Waltham, MA, USA). The cells were then washed, stained with 4,6-diamidino-2-phenylindole (DAPI) and mounted with Fluoromount G medium. Staining was observed and recorded with an inverted laser confocal microscope Axio Observer ZEISS (Germany).

### 4.10. Processing Samples for Histological Analysis Procedures

Kidney samples were fixed in 4% paraformaldehyde for 48 h and embedded in paraffin. Slices (3 mm) were stained.

Immunohistochemical analyses of E-cadherin and α-SMA were performed after 20 min of citrate unmasking at 95 °C. BOND polymer refines detection system was used in an automated Leica BOND-III system (both from Leica, Germany) for the detection of antigen-bond primary antibodies. The stained slides were scanned with the Leica Aperio Versa system (20×) and analyzed with the Leica Aperio ImageScope 12.4.

### 4.11. Quantitative Real-Time PCR (qRT-PCR)

Total RNA was extracted with Trizol^®^ reagent and reverse transcribed using the iScript gDNA Clear cDNA Synthesis Kit from BioRad (Madrid, Spain). qPCR was performed with iTaq Universal probes Supermix from BioRad (Madrid, Spain) in a CFX Connect Real-Time System from BioRad (Madrid, Spain). We performed the analysis of relative gene expression data using the 2^−ΔΔCT^ method. Appendix A shows the primer-probe sets for mice and rats. The fold change relative to the basal condition is shown and the results are mean ± S.E.M.

### 4.12. siRNA Transfection

For siRNA knockdown experiments on MCT cell line, specific mixes of four siRNA On-Target *plus* SMARTpool (Dharmacon, Lafayette, CO, USA) were transfected using the manufacturer’s protocol with DharmaFECT1 transfection reagent (Dharmacon, Lafayette, CO, USA). 5 nM of IL-18 (L-046634-00-0005) or a non-targeting scrambled sequence (D-001810-10-05) siRNAs (as a negative control) were transfected into the cells previous to (*S_S_*)-DS-ONJ treatment and/or CKs stimuli. After 48 h of culture, the efficiency of siRNAs to silence target gene expression was determined by real-time quantitative PCR.

### 4.13. Statistical Analysis

Densitometry of the Western blots was performed using the ImageJ program. Values in all graphs represented the mean ± SEM. Statistical tests were performed using the GraphPad Prisma program (GraphPad Software, San Diego, CA, USA). Data were analyzed by one-way ANOVA, or two-way ANOVA followed by Bonferroni *t*-test or by Mann–Whitney test when comparisons were among two groups. Differences were considered significant at *p* ≤ 0.05.

## Figures and Tables

**Figure 1 ijms-23-08450-f001:**
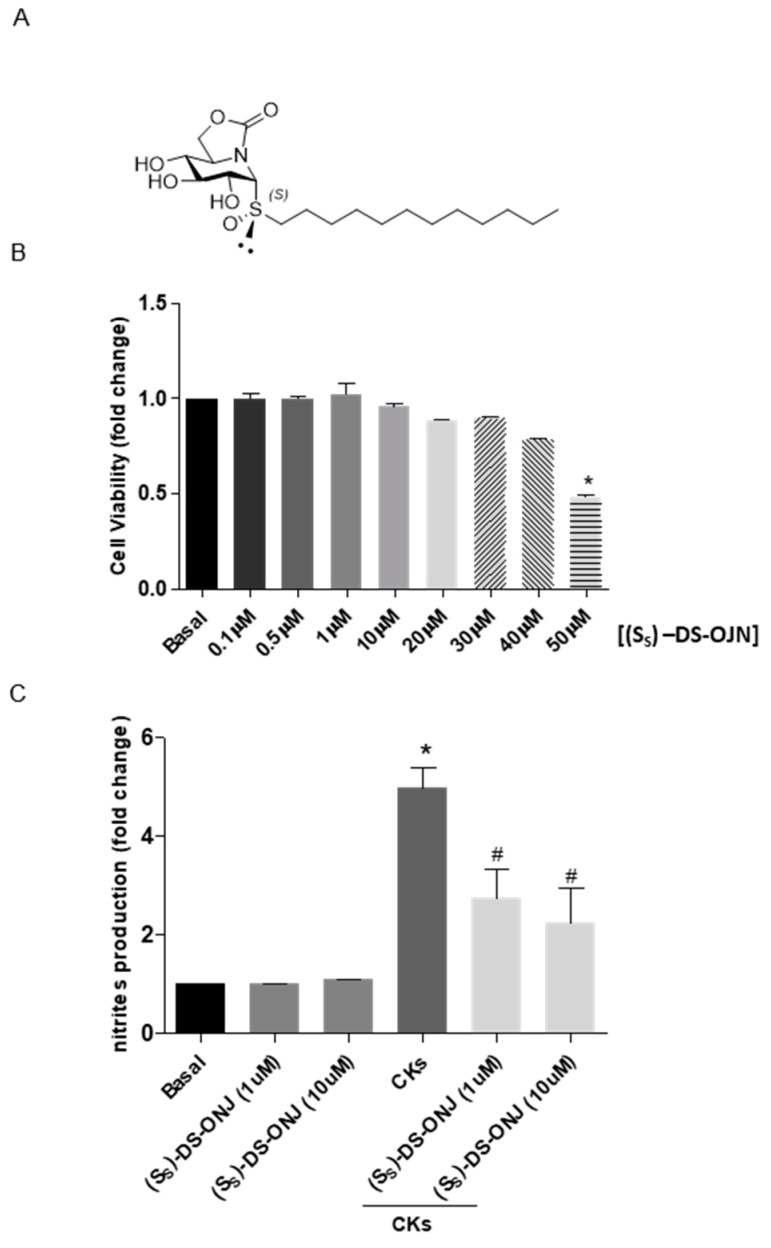
Anti-inflammatory effects of (S_S_)-DS-ONJ in CKs-stimulated MCT cells. (**A**) Chemical structure of (*S_S_*)-DS-ONJ. (**B**) MCT cells were treated for 24 h with (*S_S_*)-DS-ONJ (0.1–50 µM). Viability was determined by crystal violet staining. Colorimetric quantification was performed and the results are shown as mean ± SEM (n = 3 independent experiments). (**C**) MCT cells were treated for 24 h with CKs or CKs plus (*S_S_*)-DS-ONJ (1–10 μM). Nitrite accumulation was analyzed and related to the basal levels. Colorimetric quantification was performed, and the results are expressed as mean ± SEM (n = 6 independent experiments). The fold change relative to the basal condition is shown; * *p* ≤ 0.05 vs. Basal; ^#^
*p* ≤ 0.05 vs. CKs (two-way ANOVA followed by Bonferroni *t*-test).

**Figure 2 ijms-23-08450-f002:**
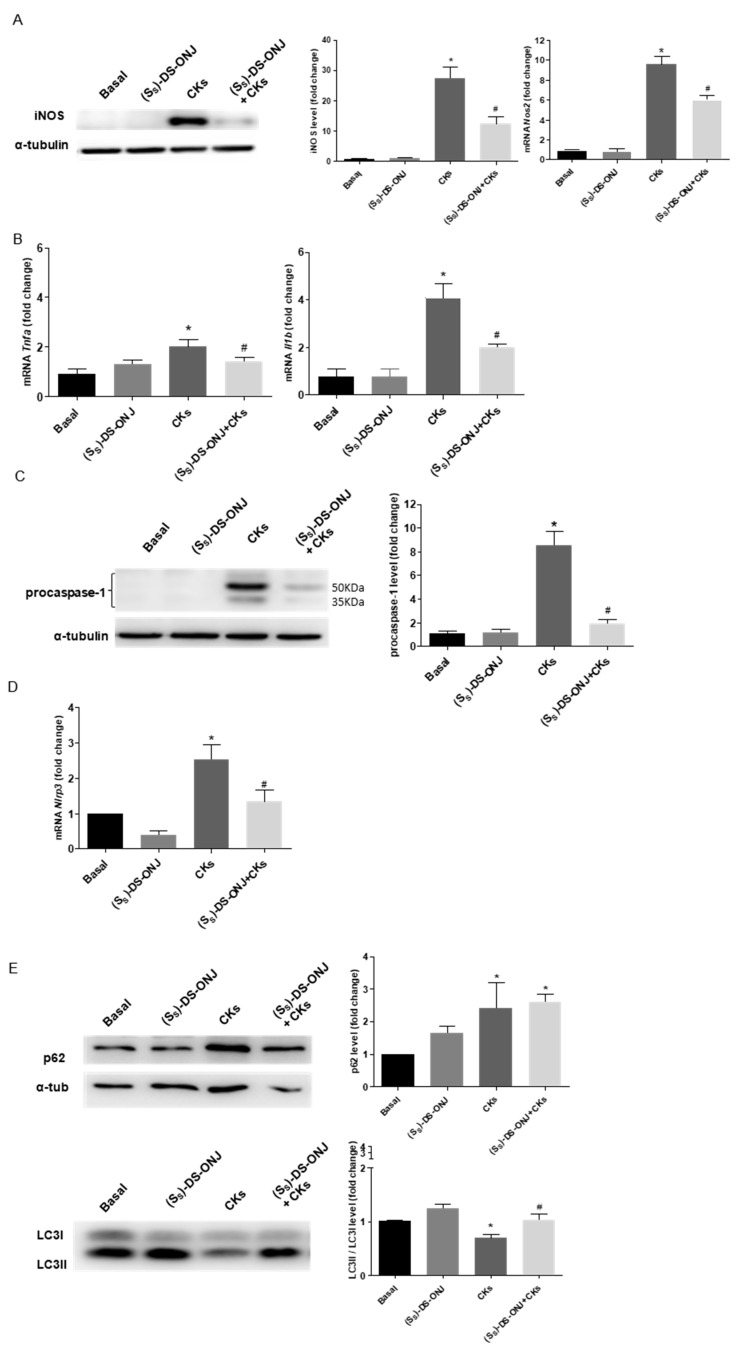
Resolution of inflammation in CKs-stimulated MCT cells by induction of autophagy flux in the presence of (S_S_)-DS-ONJ. (**A**) Protein extracts were analyzed by Western blot with the corresponding antibodies against iNOS and α-tubulin as a loading control. (n = 6 independent experiments). mRNA levels of *Nos2* were determined by qRT-PCR. The results are expressed as means ± SEM. (n = 6 independent experiments performed in triplicate). (**B**) mRNA levels of *Tnfa* and *Il1b* were determined by qRT-PCR. (n = 3 independent experiments performed in triplicate). (**C**) Protein extracts were analyzed by Western blot with antibodies against caspase-1 and α-tubulin as a loading control. (n = 6 independent experiments). (**D**) *Nlrp3* mRNA levels. *Actinb* as a housekeeping control was determined by qRT-PCR (n = 3 independent experiments performed in triplicate). (**E**,**F**) Protein extracts were analyzed by Western blot with antibodies against LC3II/I, p62 and α-tubulin as loading control for p62. (n = 4 independent experiments). The fold change relative to the basal condition is shown; * *p* ≤ 0.05 vs. Basal condition; ^#^
*p* ≤ 0.05 vs. CKs stimuli; ^+^
*p* ≤ 0.05 vs. CLQ stimuli (two-way ANOVA followed by Bonferroni *t*-test).

**Figure 3 ijms-23-08450-f003:**
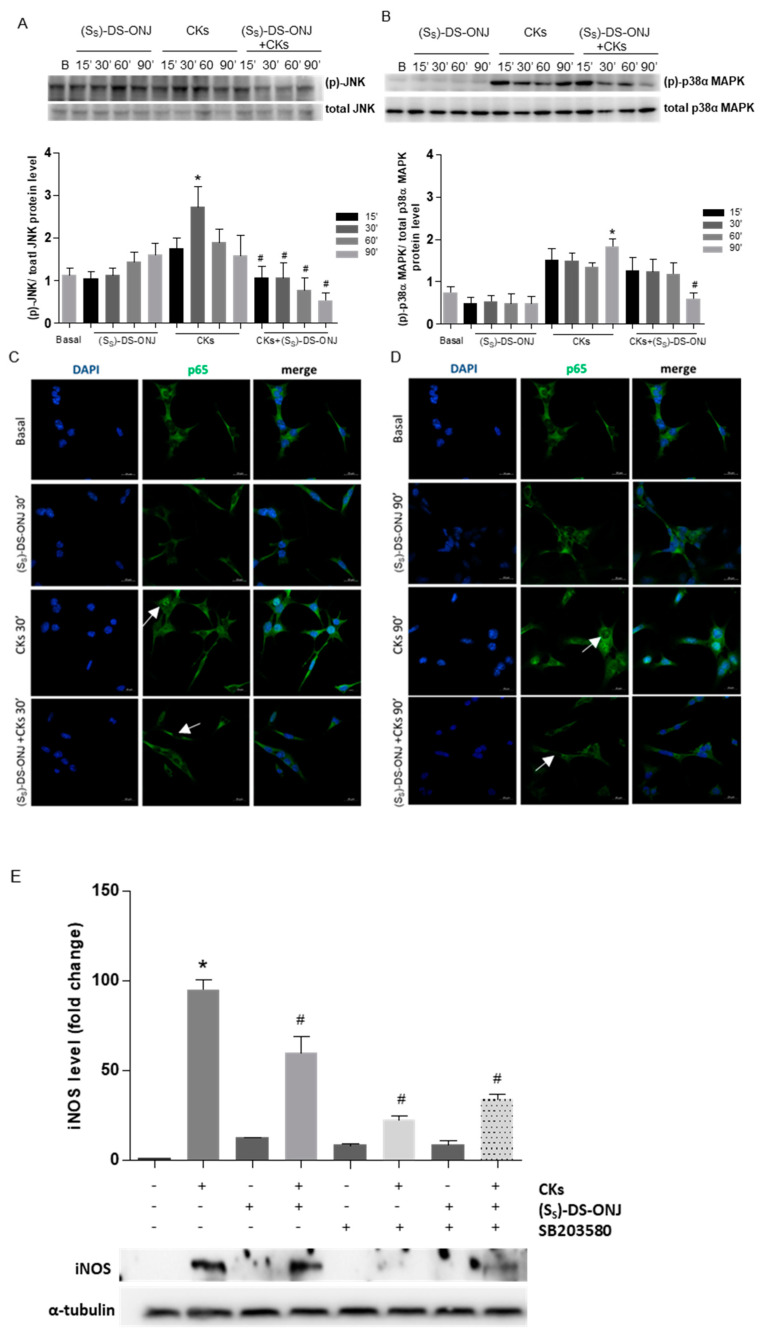
Effects of *(S_S_*)-DS-ONJ on stress kinase pathways in CKs-stimulated MCT cells. (**A**,**B**) Protein extracts were separated by SDS-PAGE and analyzed by Western blot with antibodies against phosphorylated (p)-JNK, total JNK, phosphorylated (p)-p38α MAPK and total p38α MAPK. Representative images are shown (n = 6 independent experiments). (**C**,**D**) Confocal immunofluorescence assessment of the nuclear translocation of p65-NF*κ*B in MCT cells following stimulation with CKs in the absence or presence of (*S_S_*)-DS-ONJ. The activation of p65-NF*κ*B nuclear translocation was defined as an increase in the immunofluorescence of p65-NF*κ*B (green channel) in the nuclear regions. The nuclear regions of MCT cells were visualized by counterstaining of nuclear DNA with DAPI (blue channel). The nuclear localization of p65-NFκB was detected in MCT cells upon stimulation with CKs at 30 min and 90 min, but not in MCT cells pre-treated with 10 μM (*S_S_*)-DS-ONJ (white arrows). (**E**) Protein extracts were separated by SDS-PAGE and analyzed by Western blot with antibodies against iNOS and α-tubulin as a loading control. The fold change relative to the basal condition is shown; * *p* ≤ 0.05 vs. Basal condition; ^#^
*p* ≤ 0.05 vs. CKs stimuli (two-way ANOVA followed by Bonferroni *t*-test).

**Figure 4 ijms-23-08450-f004:**
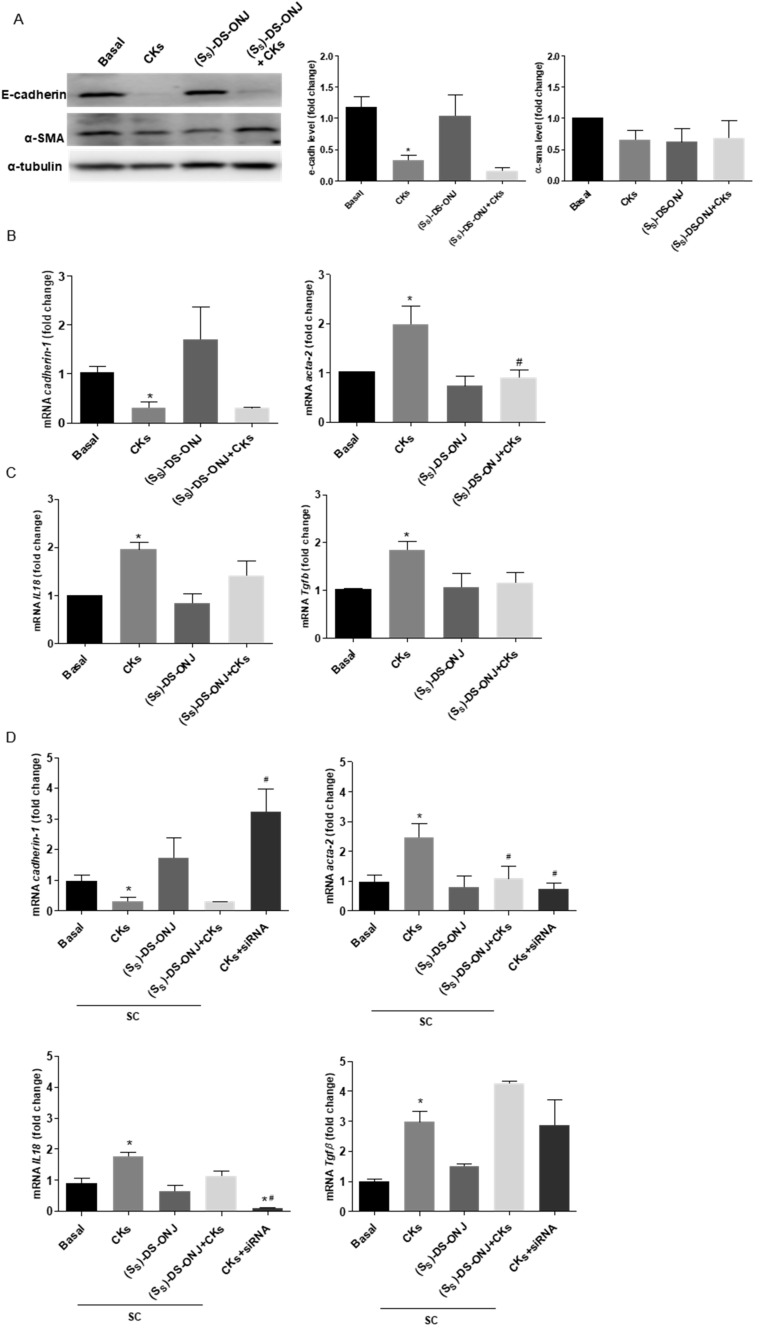
EMT progression is inhibited by (*S_S_*)-DS-ONJ in CKs-stimulated MCT cells. (**A**) Protein extracts were analyzed by Western blot with antibodies against E-cadherin, α- SMA and α-tubulin as a loading control. (n = 4 independent experiments). (**B**,**C**) *Cadherin-1*, *Acta-2*, *Tgfb* and *Il18* mRNA levels were determined by qRT-PCR. (n = 4 independent experiments performed in triplicate). (**D**) *Il18*, *Tgfb*, *Cadherin-1* and *Acta-2* mRNA levels were determined by qRT-PCR after siRNA *Il18* treatment and *Actinb* was used as housekeeping control (n = 4 independent experiments performed in triplicate). (**E**) Protein extracts were analyzed by Western blot with antibodies against ZEB-1, Snail, Slug and α-tubulin as a loading control. The fold change relative to the basal condition is shown; * *p* ≤ 0.05 vs. Basal condition; ^#^
*p* ≤ 0.05 vs. CKs stimuli (two-way ANOVA followed by Bonferroni *t*-test).

**Figure 5 ijms-23-08450-f005:**
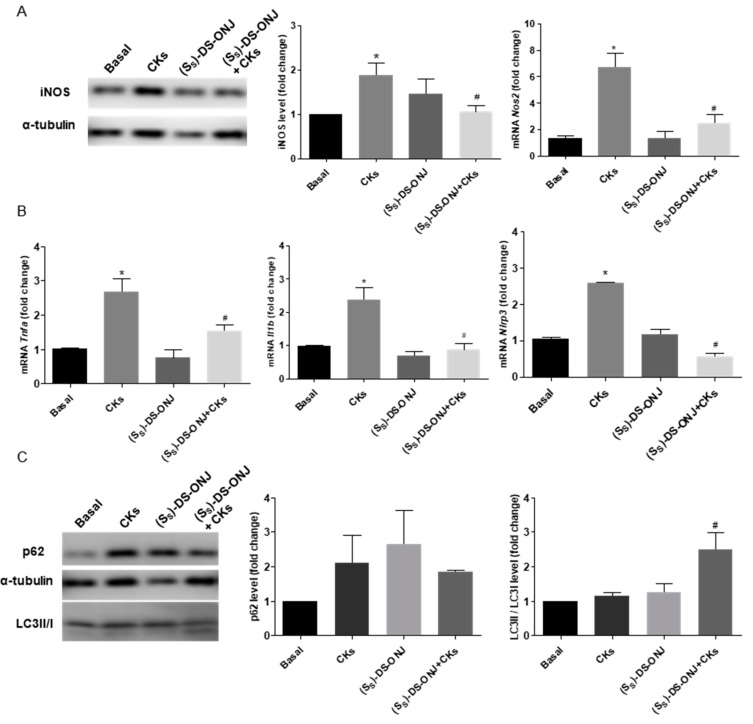
**(***S_S_*)-DS-ONJ inhibits the inflammatory events in CKs-stimulated AK explants from WT rats. (**A**) Protein extracts were analyzed by Western blot with the corresponding antibodies against iNOS and α-tubulin as a loading control (n = 6 AK explants per condition). mRNA levels of *Nos2* were determined by qRT-PCR (n = 6 AK explants per condition). (**B**) mRNA levels of *Tnfa*, *Il1b* and *Nlrp3* were determined by qRT-PCR and *Actinb* was used as housekeeping control (n = 6 AK explants per condition). (**C**) Protein extracts were analyzed by Western blot with antibodies against LC3II/I, p62 and α-tubulin as loading control of p62 (n = 6 AK explants per condition). The fold change relative to the basal condition is shown; * *p* ≤ 0.05 vs. Basal condition; ^#^
*p* ≤ 0.05 vs. CKs stimuli (two-way ANOVA followed by Bonferroni *t*-test).

**Figure 6 ijms-23-08450-f006:**
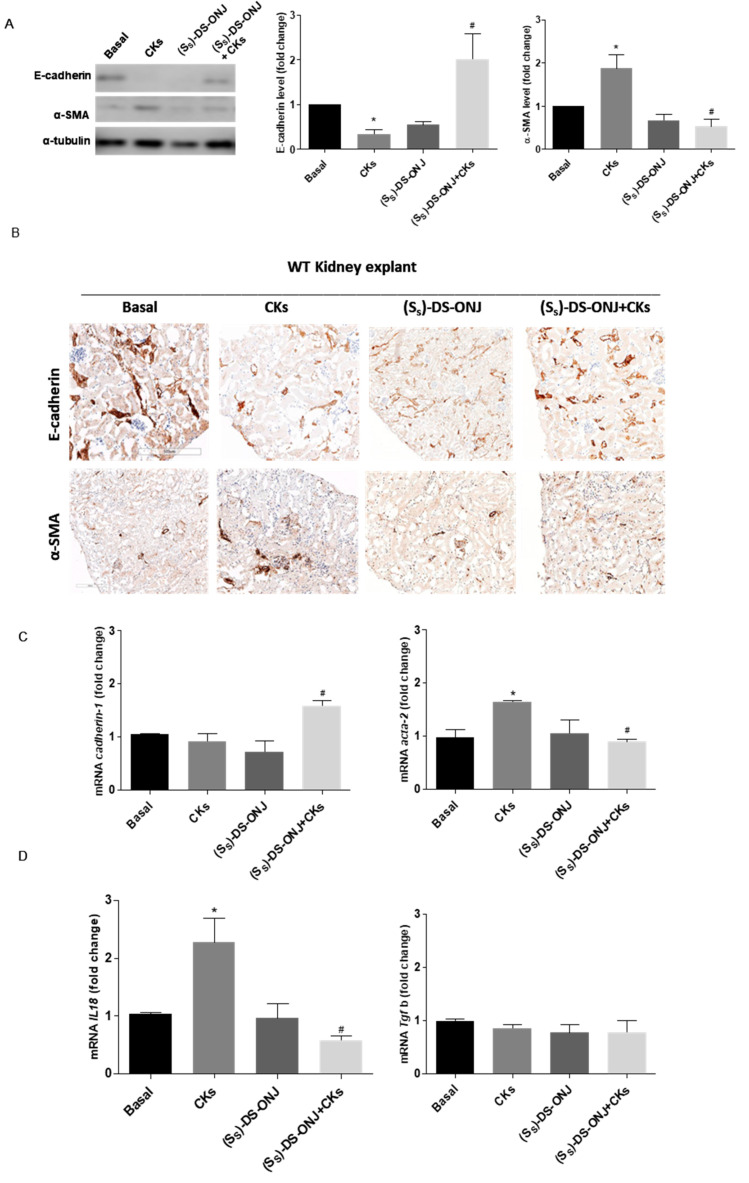
(*S_S_*)-DS-ONJ reverts the EMT induced in CKs-stimulated AK explants from WT rats. (**A**) Protein extracts were analyzed by Western blot with antibodies against E-cadherin, α-SMA and α-tubulin as loading control (n = 6 AK explants per condition). (**B**) Immunohistochemical analyses of E-cadherin (upper panel) and α-SMA (bottom panel). Original magnification: 20×. Scale bar = 0.5 mm; n = 4 AK explants per condition. Three individual slides from six different animals were analyzed. (**C**) *Tgfb*, *Cadherin-1* and *Acta-2* mRNA levels were determined by qRT-PCR and *Actinb* was used as housekeeping control (n = 6 AK explants per condition). (**D**) *Il18* mRNA levels were determined by qRT-PCR and *Actinb* was used as housekeeping control (n = 6 AK explants per condition). The fold change relative to the basal condition is shown; * *p* ≤ 0.05 vs. Basal condition; ^#^
*p* ≤ 0.05 vs. CKs stimuli (two-way ANOVA followed by Bonferroni *t*-test).

**Figure 7 ijms-23-08450-f007:**
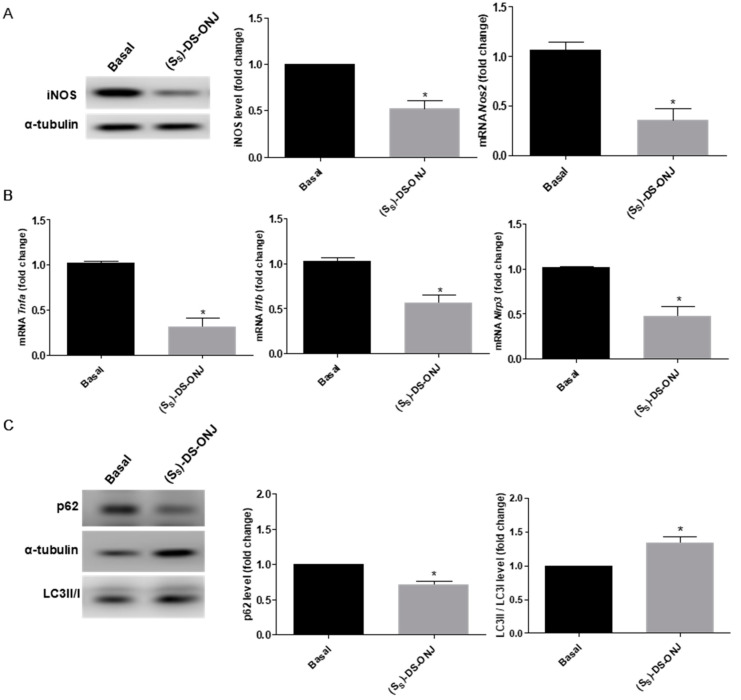
**(***S_S_*)-DS-ONJ ameliorates inflammatory parameters in AK explants from pre-diabetic BB rats. (**A**) Analysis of iNOS in AK explants from BB rats during DN progression. Protein levels were analyzed in AK explants by Western blot. α-tubulin was used as a loading control (n = 6 AK explants per condition). mRNA levels of *Nos2* were determined by qRT-PCR and *Actinb* was used as housekeeping control (n = 6 AK explants per condition). (**B**) mRNA levels of *Tnfa*, *Il1b* and *Nlrp3* were determined by qRT-PCR and *Actinb* was used as housekeeping control (n = 6 AK explants per condition). (**C**) Protein extracts were analyzed by Western blot with antibodies against LC3II/I, p62 and α-tubulin as loading control of p62 (n = 6 AK explants per condition). The fold change relative to the basal condition is shown; * *p* ≤ 0.05 vs. Basal condition (Mann-Whitney test).

**Figure 8 ijms-23-08450-f008:**
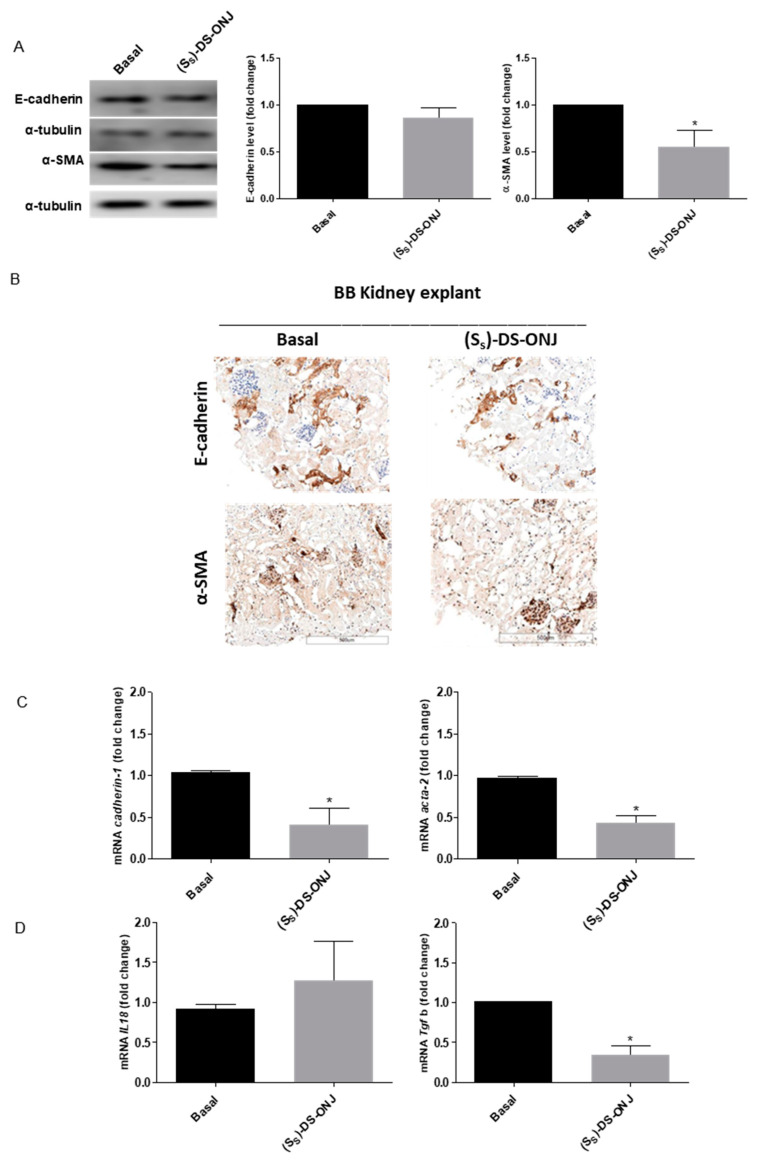
The EMT status in AK explants from pre-diabetic BB rats is not reverted by (*S_S_*)-DS-ONJ. (**A**) Protein extracts were analyzed by Western blot with antibodies against E-cadherin, α-SMA and α-tubulin as a loading control (n = 6 AK explants per condition). (**B**) Immunohistochemical analyses of E-cadherin (upper panel) and α-SMA (bottom panel). Original magnification: 20×. Scale bar = 0.5 mm; n = 4 AK explants per condition. Three individual slides from six different animals were analyzed. (**C***) Tgfb*, *Cadherin-1* and *Acta-2* mRNA levels were determined by qRT-PCR and *Actinb* was used as housekeeping control. (**D**) *Il18* mRNA levels were determined by qRT-PCR and *Actinb* was used as housekeeping control (n = 6 AK explants per condition). Three individual slides from six different animals were analyzed. (**E**) Protein extracts were analyzed by Western blot with antibodies against ZEB-1, snail and α-tubulin as a loading control. (n = 6 AK explants per condition). The fold change relative to the basal condition is shown; * *p* ≤ 0.05 vs. Basal condition (Mann–Whitney test).

## Data Availability

The dataset generated during the present study is available upon reasonable request to the corresponding authors (Ana I. Arroba and Laura Gómez-Jaramillo).

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
