# Peer review of "Unravelling the Inflammatory Processes in the Early Stages of Diabetic Nephropathy and the Potential Effect of (Ss)-DS-ONJ"

_ijms, 2022, doi:10.3390/ijms23158450_

Round 1

Reviewer 1 Report

The authors demonstrate that the glycolipid mimetic (Ss)-DS-ONJ reduce the pro-inflammatory response by autophagy flux induction in ex vivo and in vitro models.

The authors show a lot of data that (Ss)-DS-ONJ partially ameliorates inflammation, EMT, and autophagy defects caused by CK-stimulation. However, the causal relationship between inflammation, EMT, and autophagy deficiency has not been proven. It is necessary to show or mention at which pathway the protective drug effect of (Ss)-DS-ONJ might be cancelled by using knockdown approach or using specific inhibitors.

Reviewer 2 Report

The work seems excellent in its design, methodology, presentation of results and discussion.

In relation to the discussion, the authors should consider the following: extrapolate the findings obtained in the studies with explants to those described by other authors in ex vivo models of streptozotocin-dependent diabetes. At least for mechanisms related to renal fibrosis.

Round 2

Reviewer 1 Report

The comments on the peer reviewers are acceptable.

This manuscript is a resubmission of an earlier submission. The following is a list of the peer review reports and author responses from that submission.

Round 1

Reviewer 1 Report

Jaramilo et al. are assessing multiple cellular processes in a model of nephropathy and provide evidence that these pathways can be disrupted by co-incubation with (Ss)-DS-ONJ. My main critique is that the manuscript is not focused at all, authors are jumping from one pathway to another and do not provide substantial evidence that either pathway is physiologically relevant. I would recommend focusing on a single pathway, i.e. inflammasome, and describing its function in diabetic nephropathy in greater detail, testing this in additional models.

The following comments should be addressed to further recommend this manuscript for publication:

  1. Authors synthetized the compound of their interest in-house. Appropriate purity analysis against known/analytical standard should be shown in supplementary information.

  1. Response to (Ss)-DS-ONJ. There is no obvious dose response shown in Figure 1B. This sentence should be either corrected, or proper dose response curve from new analysis, revealing slope between 10 and 50 uM should performed.

  1. Experimental model. Performing experiments on one established cell lines appears to be insufficient to reach all conclusions presented in this manuscript. There are organoid lines available, as well as the option to isolate primary tubules from rodents. Is there a reason not to take advantage of further models?

  1. Autophagy & stress kinases. These phenotypes do not appear to be strong and convincing when analyzed in bulk. Reporter constructs should be used and quantification at single cell live should be performed. Immunofluorescence is not quantified. Levels of phosphor-proteins should be quantified in extended times (24 hours perhaps) and with more sensitive approach (ELISA, MS). Functional experiments exploring the actual impact of autophagy and stress kinases should be performed.

  1. Again, functional experiments that show the relevance of this finding should be shown. This is actually a nice phenotype that really deserves to be reproduced in further models and molecularly dissected.

  1. Cytokines should be quantified at protein levels. mRNA analysis is not sufficient.

  1. As per recent consensus statement, analysis of multiple EMT markers and mediators is necessary to reach such conclusion. Please, provide more evidence, ideally at protein level.

  1. AK explains. Provide histological evidence and source of the transcripts shown in Figure 5.

  1. Corresponding author email address. Both corresponding authors are listing their Gmail addresses. Institutional ones should be listed instead.

  1. Manuscript contains numerous grammar errors and has to be proofread by a native speaker.

  1. Given the model used, authors do not really determine the inflammatory stages of diabetic nephrophathy. The manuscript focuses on the effect of their compound on the MCT cell lines. Unless further, more reliable models are provided, manuscript title should be adjusted to reflect the data.

Reviewer 2 Report

The descriptive study “Unravelling the inflammatory processes in early stages of Diabetic Nephropathy and the potential effect of (Ss)-DS-ONJ” by Gómez-Jaramillo et al. addresses the protective role of glycolipid (Ss) -DS-ONJ in the origin of diabetic nephropathy (DN). For this purpose, they perform in vitro experiments with MCT cell line and ex vivo with explants of kidney cells from Bio-Breeding rats as a model of type 1 diabetes. To reproduce pre-diabetic inflammatory conditions, they use only a cocktail of cytokines. Overall, the experiments were well designed and the results are interesting and contribute significantly to our understanding of the pharmacological action of (Ss) -DS-ONJ in the early stages of DN.

However, I have some comments that need to be addressed:

  • In figure 2E the changes observed in the histogram of p62 are not appreciated in the representative image of WB. All conditions, except basal, appear the same.
  • Again in Figure 2E, how do you explain that incubation with (Ss)-DS-ONJ under basal conditions reduces LC3 I levels without increasing LC3 II levels? Furthermore, this change is not observed in the ratio of the histogram. On the other hand, the text does not read well in the graph.
  • In Figure 3A the WB loading order is different from the histogram order and confuses in the interpretation of the results. I suggest that they should have the same order in the WB as indicated in the histogram to better appreciate the changes at different times. For example, in JNK phosphorylation at 30 minutes it cannot be observed correctly since the changes are not very drastic.
  • Regarding section 2.3. from the results, observing the images, it is not appreciated that the treatment with cytokines increases the levels of alpha SMA, so it is not possible to speak of EMT in this model. Furthermore, treatment with (Ss)-DS-ONJ does not reverse mechanisms activated by CKs (Tgf-beta, IL-18). On the other hand, adding a silencing of IL-18 does not provide new information in this field.
  • In section 2.4. from the results, although the anti-inflammatory effect of (Ss)-DS-ONJ is clearly appreciated, no changes are observed in autophagy even in treatment with CKs.
  • Finally, in section 2.5. control group should have been used for comparisons. In addition, and being the strongest point in terms of criticism of the work, I think that the experiment with (Ss)-DS-ONJ in BB rats should have been carried out in vivo. Why has the treatment not been carried out directly on the experimental animals? Perhaps it is the most questionable point of the work and that it would reinforce the use of (Ss)-DS-ONJ as therapy in patients.